# Signal Detection for Ambient Backscatter Communication with OFDM Carriers

**DOI:** 10.3390/s19030517

**Published:** 2019-01-26

**Authors:** Thu L. N. Nguyen, Yoan Shin, Jin Young Kim, Dong In Kim

**Affiliations:** 1School of Electronic Engineering, Soongsil University, Seoul 06978, Korea; thunguyen@ssu.ac.kr; 2Department of Wireless Communications Engineering, Kwangwoon University, Seoul 01897, Korea; jinyoung@kw.ac.kr; 3College of Information and Communication Engineering, Sungkyunkwan University, Suwon 16419, Gyeonggi-do, Korea; dikim@skku.ac.kr

**Keywords:** ambient backscatter communication, energy detector, OFDM, test statistic

## Abstract

Ambient backscatter communication (AmBC) is considered as a promising future emerging technology. Several works on AmBC have been proposed thanks to its convenience and low cost property. This paper focuses on finding the optimal energy detector at the receiver side and estimating the corresponding bit error rate for the communication system utilizing the AmBC. Through theoretical and numerical analyses, we present two important results. First, we improve the existing energy detector by calculating the optimal averaging power orders. Second, we take advantage of the early work on orthogonal frequency division multiplexing (OFDM), where the repeating structure of ambient OFDM signals is exploited to cancel out the direct-link interference by using a cyclic prefix, then provide a test statistic in which optimal detection threshold and optimal power order are derived accordingly. The study reveals the inherent limitation of AmBC energy detectors and provides a guidance for achieving optimal power order for a given significance level.

## 1. Introduction

Ambient backscatter communication (AmBC) is a new mechanism in which a device can communicate with others by backscattering ambient radio-frequency (RF) signals (e.g, WiFi, TV signals) without any additional power suppliers [1,2]. In traditional backscatter communication such as radio-frequency identification (RFID system), a device conveys the data by modulating its reflections of an incident RF signal, which takes an expensive process for generating radio waves. For instance, a reader generates a continuous carrier wave then broadcasts it. A tag receives the signal and modulates it, and backscatters to the reader. Thus, the backscattered signal has a long delay and additional path loss. Moreover, as communicating and computing devices become smaller and abundant, powering them becomes more difficult because they require more batteries, cost, and recharging/replacement that is impractical at large scales. The AmBC solves this problem by utilizing existing RF signals, rather than generating their own radio waves. Since the RF signals are reused, the AmBC is more power-efficient and much cheaper than the traditional radio communication [3,4,5]. Therefore, the AmBC is the key building block that enables internet-of-things (IoT) and ubiquitous communication among devices with cheap and nearly zero maintenance.

Basically, RF-power devices employing the AmBC must face three main challenging issues [5,6]. First, since the backscatter signals are weak, the problem of signal detection with small changes needs to be investigated. Second, traditional backscatter receivers are constructed from powered components (e.g., oscillators), while the AmBC ones use already available RF sources, thereby reducing extra hardware cost and power consumption. In terms of spectrum sensing, the spectrum resource utilization can be much more improved since it does not need to allocate a new frequency spectrum [7]. For instance, by taking the advantage of existing RF signals in the air, it does not required any additional deployment like the RFID reader that suffers more installation and maintenance costs. The question is either how to build a network that enables ambient backscattering or build new complex digital signal processing techniques. Third, how to operate a distributed multiple access protocol and supporting the functionalities required for the AmBC should be considered. In this paper, we attempt to solve the first problem, which is the design of reader detector to recover the tag bits. There are several related works on this topic such as [5,8,9]. In [5], the authors first performed energy detection without the ability to directly measure the energy on the medium. The key insight is that if the transmitter backscatter information at a lower rate than the ambient signals, then one is able to design a receiver that can separate the two signals by leveraging the difference in communication rates. Thus, the results are very low signal-to-noise ratio (SNR) decoding and low data rate. In [8,9], the authors focused on the uplink signal detection of the communication systems adopting the AmBC, where the detectors exploit maximum a *posteriori* probability (MAP) and maximum-likelihood (ML) estimators at the receiver side. However, the solutions do not perform well when the difference between the backscatter channel and the direct-link channel is small. Other approaches [10,11] make use of WiFi backscatter to decode the tag bits by detecting the changes in the received signal strength which highly depend on channel and multi-path effects. Recently, a new AmBC over orthogonal frequency division multiplexing (OFDM) signals was proposed in [12], where the system model for such AmBC system from spread-spectrum perspective was established. By inhibiting the effects of cyclic prefix (CP) on the ambient OFDM signals, the authors developed a test statistic that is able to invalidate the inter-symbol interference among them. An extension of [12] to the case of multi-antenna receiver was presented in [13], where the test statistic was built from a linear combination of the per-antenna test statistics.

Inspired by [12], our approach is to focus on the uplink signal detection and the performance analysis for a communication system that utilizes AmBC over ambient OFDM signals. Our main ideas and contributions are highlighted as follows. First, we introduce the system model for the AmBC over ambient OFDM carriers in the air and the test statistic for tag signal detection, which are established in [12]. Second, we design an improved energy detector by proposing an arbitrary positive power operation on the signal amplitude instead of the squaring operation given as the previous work. Numerical results demonstrate that the proposed detector with optimum power order can achieve lower bit error rate (BER) and higher data rate than those in [12].

The remainder of this paper is organized as follows. Section 2 presents the system model and the problem formulation. Section 3 analyzes the optimal energy detector design for the proposed scheme. Section 4 gives the numerical results, followed by the conclusion in Section 5.

## 2. System Model and Problem Formulation

The aim of this section is first to introduce the existing AmBC system that utilizes OFDM carriers, then to present those factors that affect the detector performance. In order to understand how to make use of both energy harvesting and backscattering, we investigate the AmBC system model where an energy harvesting tag uses the existing radio signals from ambient sources to operate itself (e.g., RF source). It produces a modulated reflection of those signals to a nearby receiver (e.g., reader). Legacy receivers employ OFDM structure, where data is transmitted in parallel on a certain number of sub-carriers of different frequencies. Thus, it leads to low power consumption on data transmissions which is very suitable for low-powered hardware or batteryless IoT devices. Moreover, in terms of efficient spectrum sensing, it has been shown that energy detection approach has low computational complexity and ability to identify the spectrum holes without a *priori* knowledge of primary characteristic [8,14,15]. Once we develop an appropriate test statistic for energy detector, it can guarantee a desired detection performance.

### 2.1. Notations

The following notations are used throughout the paper.
E(·) and Var(·) denote the expectation and variance operators, respectively.$N(\mu ,\sigma^{2})$ and $\mathrm{CN}(\mu ,\sigma^{2})$ denote the Gaussian and the circularly-symmetric Gaussian distributions with mean μ and variance $\sigma^{2}$, respectively.Re{·} is the real part of a complex number.A random variable *X* that is gamma-distributed with shape *k* and rate θ is denoted as X∼Γ(k,θ). The corresponding probability density function (PDF) in the shape-rate parametrization is $f\left(x;k,\theta \right)=\frac{1}{\theta^{k}\Gamma \left(k\right)}x^{k-1}\exp\left\{-x/\theta \right\}$, where $\Gamma \left(k\right)=\int_{0}^{\infty }x^{k-1}e^{x}dx,k\in \left(0,\infty \right)$ is the gamma function evaluated at *k*.


### 2.2. Overall System Architecture

The overall system architecture utilizing the AmBC over OFDM carriers is illustrated in Figure 1. In this system, we consider two communication components coexist: the legacy OFDM system and the AmBC system. In the legacy OFDM system, the RF source transmits OFDM signals to its legacy users, while in the AmBC system a backscatter tag transmits its modulated signals to the reader over ambient OFDM carriers from the RF source. Note that this tag is equipped with a switch that can split the received signal into two parts: information decoder (ID) and energy harvester (EH). Assume that they are all connected to a single antenna and use the same RF signals. The RF-powered passive tag communicates with the reader by switching its antenna impedance of its backscattered signals. The energy harvester collects the energy from the ambient OFDM signals and uses it to provide a small amount power required for the communication and performing tasks at the tag. Finally, the backscattered signal is received and decoded by the reader.

Mathematically, the RF source transmit a passband signal s˜(n)=RePss(n)exp{jω2πfcnfs}, where s(n) is the equivalent complex baseband signal with unit power, $P_{s}$ is the average transmit power, $f_{c}$ is the carrier frequency, and $f_{s}$ is the OFDM bandwidth. The tag receives the RF source signal and transmit its modulation signal $\tilde{c}\left(n\right)$ to the reader. When we add the CP, the ambient OFDM signals are converted to a serial form and transmitted through a wireless channel. Suppose that the channel impulse response of a multipath channel is modeled as a finite impulse response filter with a certain number of taps. We denote $N_{st},N_{tr}$, and $N_{sr}$ as the number of taps corresponding to $h_{st}\left(n\right)$, $h_{tr}\left(n\right)$, and $h_{sr}\left(n\right)$, respectively. Here, we define the maximum delay of the multipath channels as $L≜\max\{N_{sr},N_{st}+N_{tr}-1\}$. Let *N* be the number of subcarriers of OFDM signals s(n) and $N_{cp}$ be the CP length. In order to gain perfect timing alignment and frequency synchronization at the receiver side, we assume that the maximum delay spread of the channel is less than the length of the CP, i.e., $L\ll N_{cp}$. Theoretically, increasing the number of OFDM subcarriers *N* leads to larger delay spread. With a fixed available bandwidth, it may cause the frequency mismatch problem between two neighbor subcarriers because the subcarrier spacing is very small. On the other hand, the number of OFDM carriers should be proportional to the CP length in practice. Thus, a tradeoff between the CP length and the subcarrier spacing must be obtained for a reasonable design.

Let x(n) be the tag data signal, thus, the received signal at the reader is given as
(1)y(n)=[ηc(n)x(n)]htr(n)︸yb(n):receivedbackscatteredsignalfromthetag+Pss(n)hsr(n)︸yd(n):direct−linkinterferencefromtheRFsource+w(n),
where $w\left(n\right)∼\mathrm{CN}\left(0,\sigma^{2}\right)$ and η are the noise and the signal attenuation parameter inside the tag, respectively.

### 2.3. Tag Operation

To ensure the orthogonality of received subcarriers over the useful symbol period as well as efficient joint allocation of subcarriers and powers among legacy users, we need to design a waveform to convey information bit in tag symbol, where the CP is longer than the delay spread of the channel. In fact, the tag uses the waveform construction x(n) in [12] to convey the bit *B* in each tag symbol as
(2)$$x\left(n\right)=\left\{\begin{array}{cc} \Pi \left(n\right)+\Pi \left(n-\frac{N+N_{cp}}{2}\right),\; & \mathrm{if}\;B(n)=0\;(\mathrm{bit}\;0),\\ \Pi \left(n\right)-\Pi \left(n-\frac{N-N_{cp}}{2}\right),\; & \mathrm{if}\;B(n)=1\;(\mathrm{bit}\;1), \end{array}\right.$$
where the square function Π(n) is defined as
(3)$$\Pi \left(n\right)=\left\{\begin{array}{cc} 1\; & \mathrm{for}\;n=0,1,\cdots ,\frac{N+N_{cp}}{2}-1,\\ 0\;, & \mathrm{otherwise} \end{array}\right..$$


### 2.4. Received Signal at the Reader

At the reader, due to the multipath effect, two portions of the direct-link interference signal $y_{d}\left(n\right)$ in each OFDM symbol period are identical, i.e., $y_{d}\left(n\right)=y_{d}\left(n+N\right),n=N_{sr}-1,\cdots ,N_{cp}-1$. Similarly, for the received ambient OFDM signal c(n) at the tag, $c\left(n\right)=c\left(n+N\right),n=N_{st}-1,\cdots ,N_{cp}-1$. Finally, we obtain the received backscattered signal $y_{b}\left(n\right)$ at the reader as
(4)$$y_{b}\left(n\right)=\left\{\begin{array}{cc} y_{b}\left(n+N\right),\; & \mathrm{if}\;B=0\\ -y_{b}\left(n+N\right),\; & \mathrm{if}\;B=1 \end{array}\right..$$


For $n=L-1,\cdots ,N_{cp}-1$, we have
(5)$$z\left(n\right)≜y\left(n\right)-y\left(n+N\right)=\left\{\begin{array}{cc} v(n),\; & \mathrm{if}\;B=0\\ u(n)+v(n),\; & \mathrm{if}\;B=1 \end{array}\right..$$


Here, $v\left(n\right)∼\mathrm{CN}\left(0,\sigma_{v}^{2}\right)$ with $\sigma_{v}^{2}=\rho \sigma^{2}$ where ρ is the noise uncertainty factor with a given upper bound noise uncertainty (in dB) $B=\mathrm{supp}\{10\log_{10}\rho \}$ [8], and $u\left(n\right)∼\mathrm{CN}\left(0,\sigma_{u}^{2}\right)$ with $\sigma_{u}^{2}=4P_{s}{\left|\eta \right|}^{2}|h_{tr}{|}^{2}\sum_{l=0}^{N_{st}-1}{\left|h_{st}\left(l\right)\right|}^{2}$. Our goal is to design a test statistic for the reader to recover the tag signal x(n) from the received signal y(n) without knowledge on OFDM ambient signal $\sqrt{P_{s}}s\left(n\right)$ transmitted from RF source. We begin by exterminating the following detection problem which tries to distinguish between the hypotheses $H_{0}$ and $H_{1}$.
(6)$$\begin{array}{cc} H_{0}:\;z\left(n\right)=v\left(n\right), & \mathrm{if}\;B=0\\ H_{1}:\;z\left(n\right)=u\left(n\right)+v\left(n\right), & \mathrm{if}\;B=1 \end{array}.$$


We define the detection SNR as $\gamma ≜\frac{\sigma_{u}^{2}}{\sigma_{v}^{2}}$.

## 3. Optimal Detector Design

Let $\pi_{0}=P\left(H_{0}\right)$ and $\pi_{1}=P\left(H_{1}\right)$, then $\pi_{0}+\pi_{1}=1$. Regarding to [16], a decision function δ(x) partitions the observation domain *R* into two disjoint sets $R_{0}$ and $R_{1}$, where
(7)$$R_{0}=\left\{x:\delta \left(x\right)=0\right\},\;R_{1}=\left\{x:\delta \left(x\right)=1\right\}.$$


We also observe that we have two possible incorrect decisions: (i) probability of false alarm, $P_{f}$ (type-I error) and (ii) probability of miss detection, $P_{m}\left(\delta \right)=1-P_{d}\left(\delta \right)$ (type-II error), where $P_{d}\left(\delta \right)$ is the probability of correct detection. Mathematically, we express
$$\begin{array}{cc} P_{f} & =P(H_{1}\;\mathrm{was}\;\mathrm{chosen}\;\mathrm{when}\;H_{0}\;\mathrm{true}),\\ P_{d} & =P(H_{1}\;\mathrm{was}\;\mathrm{chosen}\;\mathrm{when}\;H_{1}\;\mathrm{true}). \end{array}$$


In [16], Neyman and Pearson formulated the binary hypothesis testing problem pragmatically by selecting the test δ that maximizes $P_{d}\left(\delta \right)$ or equivalently that minimizes $P_{m}\left(\delta \right)$, while ensuring that $P_{f}\left(\delta \right)$ is less than or equal to a number α. The energy detector is derived by using the generated likelihood ratio test approach [16], where $u\left(n\right)∼\mathrm{CN}\left(0,\sigma_{u}^{2}\right)$ and $v\left(n\right)∼\mathrm{CN}\left(0,\sigma_{v}^{2}\right)$.
(8)$$L\left(x\right)=\frac{f_{1}\left(z\right)}{f_{0}\left(z\right)}\underset{H_{0}}{\overset{H_{1}}{≷}}\tau ,$$
where τ is chosen such that $P_{f}=\int_{L(z)>\tau }f\left(z|H_{0}\right)dz=\alpha$. We define z={z(n)},u={u(n)}, $v=\left\{v\left(n\right)\right\}\;\left(n=L-1,\cdots ,N_{cp}-1\right)$, and $D=N_{cp}-L+1$. For our hypotheses $H_{0}$ and $H_{1}$, the PDFs of the samples can be derived as
(9)$$\begin{array}{c} f_{0}\left(z\right)=\frac{1}{{\left(2\pi \sigma_{v}^{2}\right)}^{D/2}}\exp\left\{-\frac{\sum_{n=L-1}^{N_{cp}-1}{\left|z\left(n\right)\right|}^{2}}{2\sigma_{v}^{2}}\right\}, \end{array}$$
(10)$$\begin{array}{c} f_{1}\left(z\right)=\frac{1}{{\left(2\pi \sigma_{v}^{2}\right)}^{D/2}}\exp\left\{-\frac{\sum_{n=L-1}^{N_{cp}-1}{\left|z\left(n\right)-u\left(n\right)\right|}^{2}}{2\sigma_{v}^{2}}\right\}. \end{array}$$


Considering the same detection problem of (Equation 6), we define a new test as the following to improve the detection performance.
(11)$$t≜\frac{1}{D}\sum_{n=L-1}^{N_{cp}-1}\frac{{\left|z\left(n\right)\right|}^{p}}{\sigma_{v}^{p}}\underset{H_{0}}{\overset{H_{1}}{≷}}\tau .$$


Here, p>0 is an arbitrary constant which is discussed later, and τ is the detection threshold to be determined. Then, the test statistic follows the Gamma distribution with shape $k_{i}$ and scale $\theta_{i}$, i.e., $t∼\Gamma \left(k_{i},\theta_{i}\right)\;\mathrm{under}\;H_{i}$, where $k_{i}=\frac{{\left[E\left(t|H_{i}\right)\right]}^{2}}{\mathrm{Var}(t|H_{i})},\theta_{i}=\frac{\mathrm{Var}(t|H_{i})}{E(t|H_{i})}\;\left(i=0\;\mathrm{or}\;1\right)$. We denote $F_{0}\left(\cdot \right),F_{1}\left(\cdot \right)$ are the cumulative distribution functions (CDFs) of the Gamma variable *t* under $H_{0}$ and $H_{1}$, respectively, thus, $F_{i}\left(z;k_{i};\theta_{i}\right)=\int_{0}^{z}\frac{1}{\theta_{i}^{k_{i}}\Gamma \left(k_{i}\right)}x^{k_{i}-1}e^{-x/\theta_{i}}dx\;\left(i=0,1\right)$. Then, we have
(12)$$\begin{array}{c} P_{f}=P\left(t>\tau |H_{0}\right)=1-F_{0}\left(\tau ;k_{0},\theta_{0}\right), \end{array}$$
(13)$$\begin{array}{c} P_{d}=P\left(t>\tau |H_{1}\right)=1-F_{1}\left(\tau ;k_{1},\theta_{1}\right). \end{array}$$


To set the threshold, we set $P_{f}=\alpha$ and thus, $\tau =F_{0}^{-1}\left(1-\alpha ,k_{0},\theta_{0}\right)$, resulting in $P_{d}=1-F_{1}\left(F_{0}^{-1}\left(1-\alpha ,k_{0},\theta_{0}\right);k_{1};\theta_{1}\right)$. According to the central limit theorem (CLT) [16], as *D* becomes large we can represent $t∼N\left(E(t),\mathrm{Var}(t)\right)$. By assuming that ${\left|z\left(n\right)\right|}^{p}/\sigma_{v}^{p}$ are independent and identically distributed random variables, we obtain
(14)$$\begin{array}{c} E(t|H_{0})=\mu_{0};\;\mathrm{Var}\left(t|H_{0}\right)=\frac{\sigma_{0}^{2}}{D}, \end{array}$$
(15)$$\begin{array}{c} E(t|H_{1})=\mu_{1};\;\mathrm{Var}\left(t|H_{1}\right)=\frac{\sigma_{1}^{2}}{D}. \end{array}$$


Here, we have
(16)$$\begin{array}{c} \mu_{0}=\frac{2^{p/2}}{\sqrt{\pi }}\Gamma \left(\frac{p+1}{2}\right),\mu_{1}=\frac{2^{p/2}{\left(1+\gamma \right)}^{p/2}}{\sqrt{\pi }}\Gamma \left(\frac{p+1}{2}\right), \end{array}$$
(17)$$\begin{array}{c} \sigma_{0}^{2}=\frac{2^{p}}{\sqrt{\pi }}\left[\Gamma \left(\frac{2p+1}{2}\right)-\frac{1}{\sqrt{\pi }}\Gamma^{2}\left(\frac{p+1}{2}\right)\right], \end{array}$$
(18)$$\begin{array}{c} \sigma_{1}^{2}=\frac{2^{p}{\left(1+\gamma \right)}^{p}}{\sqrt{\pi }}\left[\Gamma \left(\frac{2p+1}{2}\right)-\frac{1}{\sqrt{\pi }}\Gamma^{2}\left(\frac{p+1}{2}\right)\right], \end{array}$$
where $\Gamma \left(k\right)=\int_{0}^{\infty }x^{k-1}e^{x}dx\;\left(k>0\right)$ is the Gamma function evaluated at *k*. Thus, the probabilities of false alarm and correct detection can be evaluated as
(19)$$P_{f}\approx Q\left(\frac{\tau -\mu_{0}}{\sigma_{0}/\sqrt{D}}\right),\;P_{d}\approx Q\left(\frac{\tau -\mu_{1}}{\sigma_{1}/\sqrt{D}}\right).$$


To set the threshold, we have $P_{f}=\alpha$, and thus, $\tau =Q^{-1}\left(\alpha \right)\sigma_{0}/\sqrt{D}+\mu_{0}$, resulting in
(20)$$\begin{array}{cc} P_{d} & \approx Q\left(\frac{Q^{-1}\left(\alpha \right)\sigma_{0}+\sqrt{D}\left(\mu_{0}-\mu_{1}\right)}{\sigma_{1}}\right). \end{array}$$


Thus, the overall BER is given by
(21)$$\begin{array}{cc} P_{e} & =\pi_{0}P_{f}+\pi_{1}\left(1-P_{d}\right). \end{array}$$


Considering equal probabilities of each type of error, i.e., $\pi_{0}=\pi_{1}=1/2$, the minimum value of BER is achieved by taking the derivative of $P_{e}$ with respect to τ and letting it to zeros, resulting the optimal detection threshold $\tau^{*}$. The detailed derivation is given in the Remark 1.

**Remark 1** (Optimal value of detection threshold)**.**
*In order to find the optimal value $\tau^{*}$, we need to solve the following equation*
(22)$$\begin{array}{c} \frac{1}{\sqrt{2\pi \sigma_{0}^{2}/D}}\exp\left\{-\frac{{\left(t-\mu_{0}\right)}^{2}}{2\sigma_{0}^{2}/D}\right\}=\frac{1}{\sqrt{2\pi \sigma_{1}^{2}/D}}\exp\left\{-\frac{{\left(t-\mu_{1}\right)}^{2}}{2\sigma_{1}^{2}/D}\right\}. \end{array}$$


By taking the natural logarithm on both sides, (Equation 24) can be simplified as
(23)$$\frac{D}{2}\left[\frac{1}{\sigma_{0}^{2}}-\frac{1}{\sigma_{1}^{2}}\right]t^{2}+D\left[\frac{\mu_{0}}{\sigma_{0}^{2}}-\frac{\mu_{1}}{\sigma_{1}^{2}}\right]t+\left[\ln\frac{\sigma_{0}}{\sigma_{1}}+\frac{D}{2}\left(\frac{\mu_{0}^{2}}{\sigma_{0}^{2}}-\frac{\mu_{1}^{2}}{\sigma_{1}^{2}}\right)\right]=0.$$


The above equation is a quadratic form, thus, the optimal detection threshold is given by $\tau^{*}=-\frac{\xi_{1}}{2}+\sqrt{{\left(\frac{\xi_{1}}{2}\right)}^{2}-\xi_{2}},$ where
(24)$$\frac{\xi_{1}}{2}=\left(\frac{\mu_{0}}{\sigma_{0}^{2}}-\frac{\mu_{1}}{\sigma_{1}^{2}}\right)\left(\frac{1}{\sigma_{0}^{2}}-\frac{1}{\sigma_{1}^{2}}\right),\;\xi_{2}=\frac{\frac{2}{D}\ln\frac{\sigma_{0}}{\sigma_{1}}+\left(\frac{\mu_{0}^{2}}{\sigma_{0}^{2}}-\frac{\mu_{1}^{2}}{\sigma_{1}^{2}}\right)}{\left(\frac{1}{\sigma_{0}^{2}}-\frac{1}{\sigma_{1}^{2}}\right)}.$$


**Remark 2** (Optimal value of power order *p*)**.**
*In [12], the value p is fixed at p=2. In our paper, the value p is chosen to maximize $P_{d}$ at fixed $P_{f},\gamma$, and D. Thus, the optimal value of $p^{*}$ is obtained by solving*
(25)$$\begin{array}{cc} p^{*} & =\arg\max_{p}P_{d}\\ & =\arg\max_{p}Q\left(\frac{Q^{-1}\left(\alpha \right)\sigma_{0}/\sqrt{D}+\mu_{0}-\mu_{1}}{\sigma_{1}/\sqrt{D}}\right). \end{array}$$

*If we assume $P_{d}$ to be differentiable, then we can differentiate both sides*
(26)$$\frac{\partial P_{d}}{\partial p}=0.$$

*The derivation of solving this equation is given in Appendix A.*


## 4. Numerical Results

In [12], the authors compared the energy detector with a benchmark design in which the reader detected the tag bit by distinguishing between two different orders of the average power of the received signal y(n), where
(27)$$\bar{z}≜\frac{1}{N+C_{cp}}\sum_{n=1}^{N+N_{cp}-1}{\left|y\left(n\right)\right|}^{2}.$$


They also showed that the their design was comparable to the benchmark design in terms of complexity, but the performance was better in terms of transmission rate and BER. Thus, we compare our proposed approach with [12] (i.e., p=2), which is referred as the “conventional” energy detector. A summary description of simulation parameters is given in Table 1. For comparison purpose, we keep all the parameter as the same for both detection schemes. In fact, if the tag backscatters the information at a lower rate than the ambient signals, we can design a receiver that can separate two signals [5,12].

In the following, we briefly describe some metrics that we used to evaluate the proposed statistic test. The test must be a sufficient statistic for our energy detection problem and contains all the information required to distinguish two hypotheses $H_{0}$ and $H_{1}$.
(i)First, we check the validity of the Gaussian approximation for the proposed test. In fact, since the test statistic (Equation 11) follows the Gamma distribution under both hypotheses, the length *D* must be large enough to apply the CLT while not very large to keep the approximation meaningful. In the first run, Figure 2 gives us a case study on this approximation.(ii)Second, in order to find the optimal *p* for the test (Equation 11) instead of using p=2, we solve (Equation 27) to obtain an adaptive power order. The result is shown in Figure 3. The purpose of this result is to observe how the $p^{*}$ changes for maximizing the probability of correct detection according to the changes in the SNR. Thus, we may have a certain strategy to select *p* for a given SNR and a false alarm rate.(iii)Third, with different settings (e.g., SNR and $N_{cp}$), we observe how much the BER changes when using our test statistics and the conventional ones in terms of our ability to solve $p^{*}$ with high correct detection probability $P_{d}$. The results are given in Figure 4 and Figure 5.(iv)Finally, we provide the median receiver operating characteristics (ROC) curve for our detector design as predicted by our aforementioned analysis.


Following the above construction, in order to verify the accuracy of approximating the simulated PDFs (i.e., Gamma distribution) by the theoretical approximation (i.e., *Q*-function), Figure 2 illustrates those PDFs when γ=0 dB, $N_{cp}=64$, and N=512, thus, L=8 and D=57. We observe that the Gamma approximation fits well in most cases considered. The accuracy of the approximation increases when *p* decreases. This approximation may be adequate for practical energy detectors since we can improve it by increasing the length *D* in the detection. Thus, we have to select appropriate values of length *D* and $P_{f}$ to guarantee high probability of detection, while keeping the Gaussian approximations to be valid.

Figure 3 shows the optimum value of *p* in (Equation 27) with different fixed values of $P_{f}=\alpha$. The value $p^{*}$ maximizing $P_{d}$ decreases when γ increases. We also plot a small subgraph at the right hand side of Figure 3 to illustrate the $p^{*}$ value (in vertical axis) versus small SNR (in horizontal axis) because we observe that $p^{*}$-curve has a big jump in its value for γ in the range of (0,1.5). We offer some brief comments. First, the value $p^{*}$ is probably the best solution we has achieved through numerical results. The $p^{*}$-curve grows sub-linearly versus small value of SNR (e.g, γ≤1) simultaneously, while it decays slowly and remains constant as a function of γ≥6. In a certain sense, the vector z represents the relative difference between two input signals. Due to weak backscattered signals, if it has any small component, p=2 makes them negligible. On the other hand, the optimal power order is around 1, which is more irritated by small values. For instance, when γ≤1, the proposed detector returns $p^{*}\approx 1$ rather than p=2. In this case, it prefers returning the number of non-zero values of the vector z={z(n)} rather than tolerates them.

Figure 4 demonstrates the theoretical BER performance of the conventional energy detector with p=2 and the proposed detector with optimal power order $p^{*}$. We set $N_{cp}=64$, N=512, and α=0.01. We can see that the energy detector performs worse than the proposed one because of the inaccurate Gaussian approximation. We also see that the improvement here becomes more prominent when the target probability of false alarm is smaller, as well as the achieved SNR is relatively higher. The BER becomes flat even for high SNR value. For the proposed detector, it can be found that the increasing SNR yields reduced BER, especially when SNR is small (e.g., SNR ≤2 dB). For larger SNR, the BER performance remains unchanged. This phenomenon may be caused by the strong direct-link interference [13]. Morever, since the reader tries to distinguish between two bits by taking a sufficiently large number of samples, i.e., *D*, the value of *D* we consider here is large enough to apply the CLT, thus, the probability of miss detection and the probability of false alarm are moderate, i.e., they are not changed with *D*. Consequently, the overall BER in (Equation 23) does not become much different. The detector must reach the error probabilities uniformly over a whole uncertainty set with various *D*. As SNR increases, it hits the SNR wall [17] while the required sample complexity meets our performance target.

Figure 5 depicts the curves of BER versus SNR with several value of $N_{cp}$ for the proposed detector. We set γ=5 dB and α=0.01. The BER approaches 0.5 at small $N_{cp}$ and there exist little gaps between BER curves. We observe that the BER decreases as $N_{cp}$ increases. However, the smaller $N_{cp}$ offers higher data rate from the relationship
(28)$$R_{\mathrm{tag}}=\frac{f_{s}}{\left(N+N_{cp}\right)},$$
where $R_{\mathrm{tag}}$ is the tag rate [12]. Obviously, if we fix the number of OFDM carriers *N*, $R_{\mathrm{tag}}$ decreases as the CP length increases, while the BER decreases, as illustrated in Figure 5. Thus, there exists a trade-off between the BER and the data rate $R_{\mathrm{tag}}$.

In order to assess the performance of the proposed detector, we plot the ROC curves in Figure 6, which shows the relationship between the probability of detection $P_{d}$ and the probability of false alarm $P_{f}$ for a given SNR γ. We confirm that the proposed approach provides better ROC values when γ is small, i.e., the performance gain becomes much larger in the lower SNR environment. As the SNR increases, the detection threshold must be set higher to obtain a good ROC.

## 5. Conclusions

In this paper, we have studied the signal detection for the AmBC system with OFDM carriers, while providing some key mathematical insights underlying this theory and proposing an improved energy detector with optimum power orders. Especially, in order to maximize the probability of correct detection, the power order of energy detector was chosen subject to the target probability of false alarm. The proposed detector was shown to improve energy efficiency for spectrum sharing via AmBC.

Moreover, based on the insightful results we suggest the following directions for future work.
(i)Regarding tag operation, an important direction is to come up with a model that examines the energy harvesting model and enhances the detection performance accordingly.(i)In our problem formulation, we use a simple noise uncertainty model, i.e., the variance of v(n) is assumed to be bounded by a given number *B*. This value depends only on a single value ρ, thus, it may not incorporate the RF strength and other changes in the environments. Therefore, we need to investigate other nonlinear models that relate to energy detector’s inherent noise uncertainty.(ii)We have shown that the proposed energy detector can be effective for the AmBC system with OFDM carriers. However, it needs more theoretical bounds on *D* and SNR γ along with numerical results.(iii)In the problem formulation, we assumed that the tag has two states: backscattering and non-backscattering, while in practice its antenna load may switch among three states: no reflecting, reflecting in the same phase, and reflecting in the opposite phase, resulting in a ternary signal B(n). Thus, we need to design a waveform x(n) to convey the corresponding bits.


We expect that the above future directions can contribute to the advancement of energy detection and estimation areas.

## Figures and Tables

**Figure 1 sensors-19-00517-f001:**
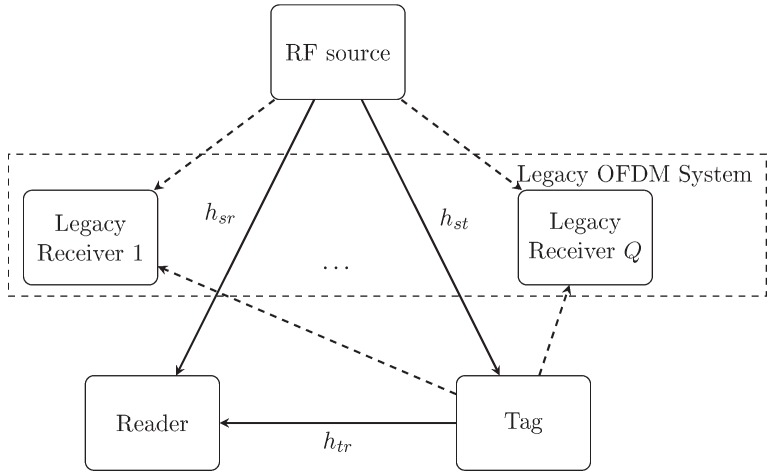
A communication system utilizing the AmBC over OFDM carriers. The AmBC system consists of three main components: RF source (e.g., TV tower), ambient backscatter transmitter (e.g., AmBC tag), and ambient backscattter receiver (e.g., reader), while the legacy OFDM system consists of several legacy receivers (e.g., mobile phones).

**Figure 2 sensors-19-00517-f002:**
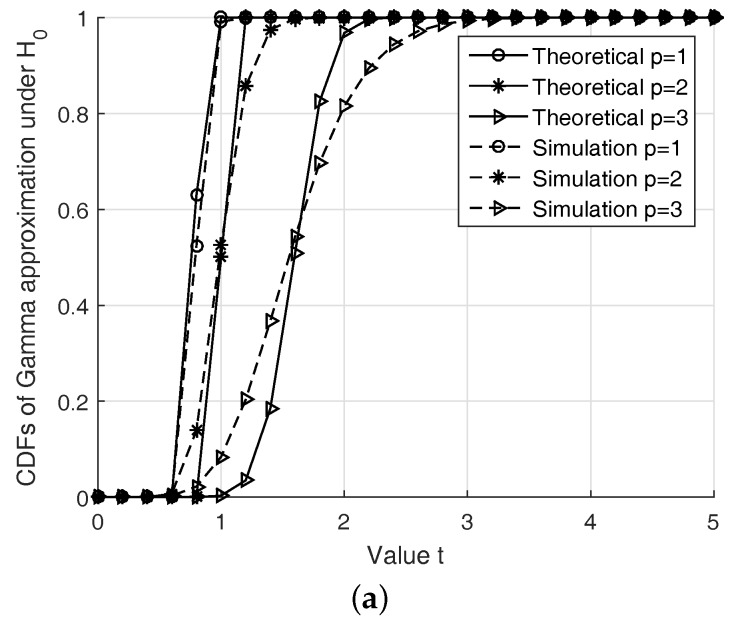
Illustration of CDFs under $H_{0}$ and $H_{1}$. Note that the theoretical analysis shows that the test statistic $t∼\Gamma \left(k_{i},\theta_{i}\right)\;\mathrm{under}\;H_{i}$, while the simulation approximation gives us t∼N(E(t),Var(t)), where E(t) and Var(t) are given in (Equation 14) and (Equation 15), respectively. (**a**) Simulated CDFs for *t* under $H_{0}$ when γ=0. (**b**) Simulated CDFs for *t* under $H_{1}$ when γ=0.

**Figure 3 sensors-19-00517-f003:**
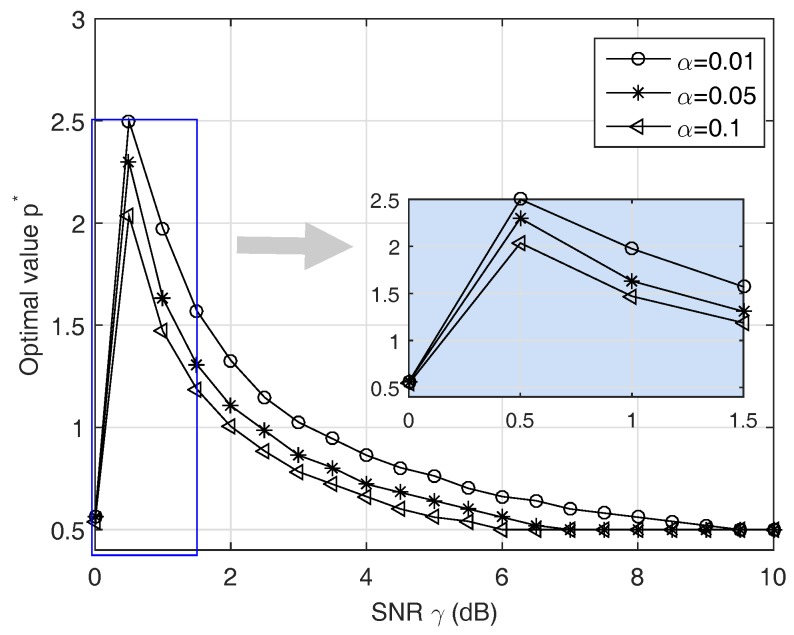
The optimum value of power order *p* versus γ. It shows the effect of γ on $p^{*}$ at several different SNR levels, which comes from solving procedure of (Equation 28). We generated an SNR vector of 21 equally-spaced points between 0 and 10. The $p^{*}$-curve is aimed to choose the appropriate *p* value for the test statistic.

**Figure 4 sensors-19-00517-f004:**
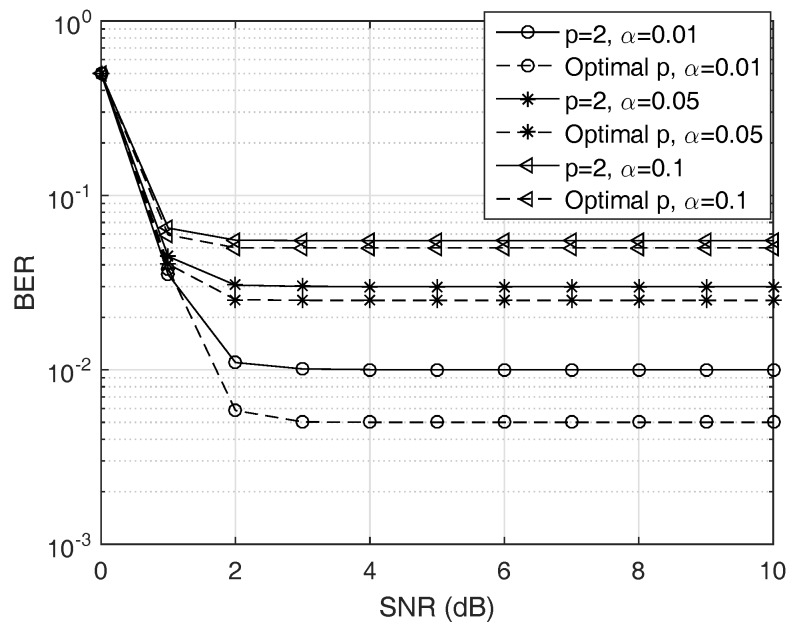
BER versus SNR γ. We observe that BER achieves the maximum at γ=0. The designed detector can perform well even when the SNR is high. In both detector schemes, the BER remains unchanged despite of large SNR because of strong direct-link interference [8] or SNR wall problem [17].

**Figure 5 sensors-19-00517-f005:**
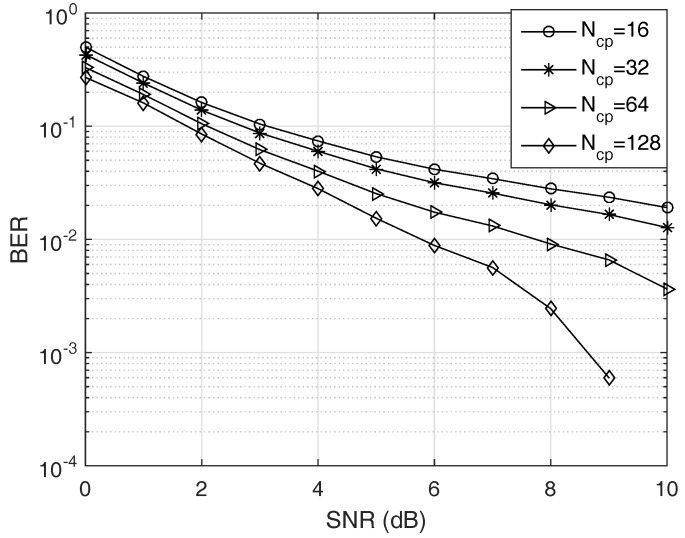
BER versus SNR with $N_{cp}$. As we predicted, the BER increases as $N_{cp}$ decreases. Since *N* is defined to be proportional to $N_{cp}$, while $N_{cp}$ should be greater than the channel delay spread *L* to eliminating the interference, there exists tradeoffs between the CP length and the subcarrier spacing as well as the CP length and the tag rate.

**Figure 6 sensors-19-00517-f006:**
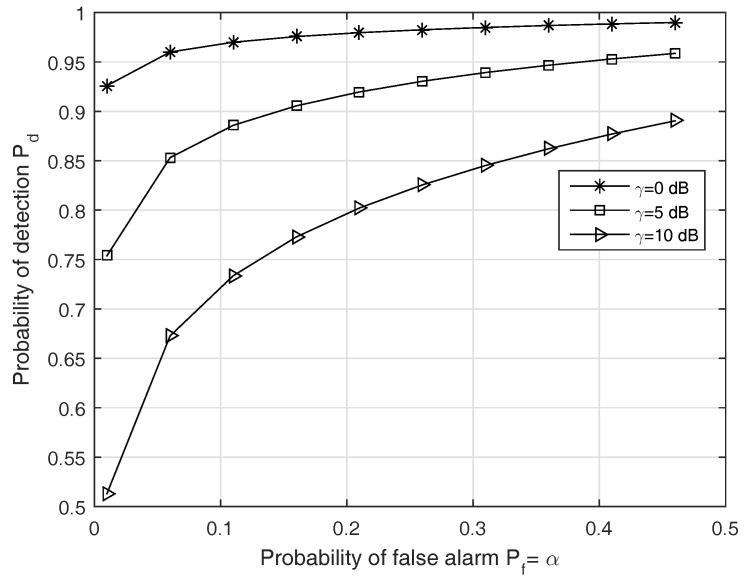
ROC curve with SNR γ. An ROC curve is obtained by taking the average over 100 independent trials. From the figure, for each false alarm rate $P_{f}$, there is a SNR value γ for achieving the objective probability of correct detection $P_{d}$.

**Table 1 sensors-19-00517-t001:** Simulation Parameters

Parameter	Value
OFDM bandwidth	$f_{s}=20$ MHz
Number of paths	$N_{sr}=N_{st}=8$, $N_{tr}=1$
Attenuation value	η=1
CP length	$N_{cp}=16,32,64,128$
Number of carriers	$N=8N_{cp}$

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
