# Peer review of "Signal Detection for Ambient Backscatter Communication with OFDM Carriers"

_sensors, 2019, doi:10.3390/s19030517_

Reviewer 1 Report

Thank you to the authors, the paper and the work are interesting! But they need to be improved. 

Some comments about the paper:

The abstract should NEVER have any references. 

The notations should have their own section, not in the end of the Introduction. 

All the sections should have their own short introduction. 

I miss a section (number 2) that covers the state of the art. The authors talk about it briefly in the introduction, but its not enough for a journal paper. I recommend them to talk about the SotA in an entire section. There is enough literature to have that section. The authors have only 12 cites in this paper, they should increase their literature review. 

In Figure 3, I suggest that tests should be done also for SNR=0.5 and 1.5 at least. The change among SNR=0, 1 or 2 is too big to interpolate it without any more tests. 

Why does the BER remain despite improving the SNR? That question is important and should be answered in the text. 

The captions of the figures should be more auto-explicative. 

The conclusions are much too short, the authors should conclude wider from their tests. 

The appendix A should have more verbal explanations. 

Author Response

Thank you for your detailed comments. The reply letter and the revised manuscript files are attached blow. Please check them out.
Thank you.

Reviewer 2 Report

 Authors, thank you for your submission. This is a very interesting topic and a thorough analysis will be helpful for other researchers.

On page 1, “Second, traditional backscatter receivers are constructed from powered components (e.g. oscillators), while the AmBC ones are battery free.” Could you add more to this statement? Do you envision the AmBC devices communicating with each other, requiring battery free operation, or with a legacy receiver or reader like in Fig. 1, where there is likely a wall connected power source or a battery?  

Page 3, “The tag receives the RF source signal and transmit its modulation signal c(n) to the reader” It would be more consistent with other backscatter works to say the tag backscatters, reflects, or modulates the RF signal, instead of saying it transmits a modulation signal.

It would be interesting to hear a discussion of the trade offs of the tag side requirements for this scheme. The tag is presumably operating from harvested energy so keeping the backscatter modulation rate low or keeping the backscatter time short is required to decrease the RF input power requirement for the tag.  How does the CP length impact the tag’s power requirement or data rate?

Please spend a little more time describing what Fig. 2 is showing. It’s clear from the written description that the theory and simulation agree well but there isn’t much description of what the plots mean.

Why are the simulation parameters in Table 1 used? How would this compare to a common OFDM carrier source?

Here are a few grammatical issues I noticed:

On page 1, “Threrefore, the AmBC is the key building block that enables internet-of-things and ubiquitous communication among devices with cheap and nearly zero maintenance.” You may want something more like, “Threrefore, AmBC is the key building block that enables internet-of-things and ubiquitous communication for cheap devices that require nearly zero maintenance.”

On page 4, “We begin by exterminating the following detection problem …” instead of exterminating you probably meant examining.

On page 6, “They also showed that the their design was comparable …” delete the extra the between that and their.

On page 7, “…while keeping the Gaussian approximations to be valid” delete “to be”

Here are a few suggested additional references:

1.     N. Parks, A. Liu, S. Gollakota, J. R. Smith, “Turbocharging Ambient Backscatter Communication”

2.     A. Wang, V. Iyer, V. Talla, J. R. Smith, and S. Gollakota, “FM backscatter: enabling connected cities and smart fabrics”

Author Response

Thank you for your detailed comments. The reply letter and the revised manuscript files are attached blow. Please check them out.
Thank you.

Round  2

Reviewer 1 Report

The authors have improved quite a lot the paper using the reviewers comments. I appreciate that effort. Nevertheless, there are still some comments to be done about that work:;

The english should be reviewed, it still has some flaws. 

Some captions of figures are still too short to be widely understood. 

I still don't agree with comment 1.5. I honlestly think that there should be more data to conclude what the authors say in the paper, more samples. 

Something similar happens with comment 1.6, that the authors detail their conclusion with more information, but do not really answer my question about why it remains unchanged. 

Thanks for your work, authors! 

Author Response

Please find the attached file for the reply letter

Reviewer 2 Report

Authors, thank you very much for addressing so many of the review comments.  With the extra explanations and detailed descriptions of figures I think the paper is much easier to follow and understand.

I appreciate your response to some of my questions but based on your response I may not have been clear enough with my comments and questions.

Comment 2.1 On page 1, “Second, traditional backscatter receivers are constructed from powered components (e.g. oscillators), while the AmBC ones are battery free.” Could you add more to this statement? Do you envision the AmBC devices communicating with each other, requiring battery free operation, or with a legacy receiver or reader like in Fig. 1, where there is likely a wall connected power source or a battery?

I know the tag isn’t generating its own radio waves and doesn’t need batteries, but I like your additional sentence clarifying that point.  My question was about where the backscatter receiver will reside.  It’s possible to have tag-to-tag communication where both tags have no batteries.  That scenario would require a backscatter receiver, a receiver that receives backscattered signals, in the tag. Only if the receiver is in the tag would you have a backscatter receiver that is battery free.  If the backscatter receiver (eg. Mobile phones) is a legacy receiver like in Fig 1. it will have a battery or be plugged into a wall. I strongly disagree with the statement, “Second, traditional backscatter receivers are constructed from powered components (e.g. oscillators), while the AmBC ones are battery free.” A battery free backscatter receiver is not the distinguishing piece of AmBC. The distinguishing piece is no custom built illuminating RF source providing a carrier wave. Please correct this statement so readers will understand the signal detection being described in the paper is for backscatter signals from a tag to either a legacy receiver or reader device.

Comment 2.5. Why are the simulation parameters in Table 1 used? How would this compare to a common OFDM carrier source?

Thank you for the clarification but what I meant was please explain to the reader why the parameters in Table 1 were chosen.  I know how the parameters were used later in the paper but why were they used? For example, are these the parameters for a WiFi signal? You should describe your thought process so the reader will understand why these parameters are reasonable choices for a simulation to evaluate your detector. Also, the value column of “OFDM bandwidth” should say “fs = 20 MHz” to be consistent with the other columns and make it easier to look up parameters in equations.  The same goes for parameter “CP length Ncp” with Value “Ncp = 16,32,64,128”

Author Response

(The authors gave the same response as above.)
